# Crystal structure of an HIV assembly and maturation switch

Jonathan M Wagner[1†], Kaneil K Zadrozny[1†], Jakub Chrustowicz[1], Michael D Purdy[1], Mark Yeager[1,2]*, Barbie K Ganser-Pornillos[1]*, Owen Pornillos[1]*

[1]Department of Molecular Physiology and Biological Physics, University of Virginia, Charlottesville, United States; [2]Department of Medicine, Division of Cardiovascular Medicine, University of Virginia Health System, Charlottesville, United States

**Abstract** Virus assembly and maturation proceed through the programmed operation of molecular switches, which trigger both local and global structural rearrangements to produce infectious particles. HIV-1 contains an assembly and maturation switch that spans the C-terminal domain (CTD) of the capsid (CA) region and the first spacer peptide (SP1) of the precursor structural protein, Gag. The crystal structure of the CTD-SP1 Gag fragment is a goblet-shaped hexamer in which the cup comprises the CTD and an ensuing type II β-turn, and the stem comprises a 6-helix bundle. The β-turn is critical for immature virus assembly and the 6-helix bundle regulates proteolysis during maturation. This bipartite character explains why the SP1 spacer is a critical element of HIV-1 Gag but is not a universal property of retroviruses. Our results also indicate that HIV-1 maturation inhibitors suppress unfolding of the CA-SP1 junction and thereby delay access of the viral protease to its substrate.

*For correspondence: my3r@ virginia.edu (MY); bpornillos@ virginia.edu (BKG-P); owp3a@ eservices.virginia.edu (OP)

†These authors contributed equally to this work

Competing interests: The authors declare that no competing interests exist.

## Introduction

Assembly and maturation of HIV-1 and other retroviruses is driven by the viral structural polyprotein, Gag, which consists of a series of independently folding domains, called MA (matrix), CA (capsid), and NC (nucleocapsid) (reviewed in *Bush and Vogt, 2014*; *Ganser-Pornillos et al., 2012*; *Lingappa et al., 2014*). During assembly, Gag makes a spherical, immature capsid shell that packages the viral genome, directs interactions with the plasma membrane to promote envelopment, and recruits cellular machinery to release the new virus particle from the host cell. The Gag shell has hexagonal paracrystalline lattice symmetry, wherein the central CA region makes two layers of interactions through its two domains (NTD and CTD, for N-terminal and C-terminal domain). During maturation, Gag is cleaved by the viral protease at specific sites. This results in disassembly of the Gag shell and release of new structural proteins, which then rearrange into the mature, infectious virion. In the mature virion, the genome is repackaged inside a cone-shaped capsid made up of the CA protein, with the NTD forming hexameric or pentameric rings that are connected by the CTD.

As with other animal viruses, the large-scale capsid rearrangements that occur during retroviral maturation are associated with the operation of structural switches that undergo dramatic conformational changes. In HIV-1, two such putative switches flank the CA region of Gag. One switch spans the MA-CA junction, which is in an extended configuration in the immature virion (*Kelly et al., 2006*; *Tang et al., 2002*). Upon proteolysis, the new CA N-terminus folds into a β-hairpin stabilized by a buried salt bridge between the new N-terminal proline and a conserved aspartate residue and promotes assembly of the mature capsid (*Gitti et al., 1996*; *von Schwedler et al., 1998*). A second putative switch spans the junction between CA and the first spacer peptide (SP1) (*Gross et al., 2000*). The CA-SP1 junction is predicted to fold into an α-helix (*Accola et al., 1998*; *Liang et al., 2002*), and forms the binding site of so-called 'maturation inhibitors,' as exemplified by 3-O-(3',3'-

**eLife digest** Viruses like HIV must undergo a process called maturation in order to successfully infect cells. Maturation involves a dramatic rearrangement in the architecture of the virus. That is to say, the virus's internal protein coat – called the capsid – must change from an immature sphere into a mature cone-shaped coat. Notably, this maturation process is what is disrupted by the protease inhibitors that are a major component of anti-HIV drug cocktails.

Structural changes in small portions of the capsid protein, termed molecular switches, commonly trigger the viral capsids to reorganize. The HIV capsid has two of these switches, and Wagner, Zadrozny et al. set out to understand how one of them – called the CA-SP1 switch – works.

Solving the three-dimensional structure of the immature form of the CA-SP1 switch revealed that it forms a well-structured bundle of six helices. This helical bundle captures another section of the capsid protein that would otherwise be cut by a viral protease. The CA-SP1 switch therefore controls how quickly the protein is cut and the start of the maturation process.

Wagner, Zadrozny et al. then discovered that other small molecule inhibitors of HIV, called maturation inhibitors, work by binding to and disrupting the transformation of the CA-SP1 switch. Finally, further experiments showed that the formation of the CA-SP1 helical bundle controls when the immature capsid shell forms and coordinates the process with the capsid gaining the genetic material of the virus.

The new structure means that researchers now know what the HIV capsid looks like at the start and end of maturation. The next challenge will be to figure out exactly how the capsid changes from one form to the next as HIV matures.

dimethylsuccinyl)betulinic acid (bevirimat) (reviewed in *Adamson et al., 2009*). These inhibitors delay proteolysis of the CA-SP1 junction and induce aberrant maturation. The importance of the CA-SP1 junction as a regulator of proteolysis during maturation is well established (*Kräusslich et al., 1995*; *Pettit et al., 1994*; *Wiegers et al., 1998*), but the molecular basis of this function has been unknown.

Mutagenesis studies also indicate that the CA-SP1 junction is important for assembly of the immature HIV-1 Gag shell (*Accola et al., 1998*; *Al-Mawsawi et al., 2014*; *Datta et al., 2011*; *Guo et al., 2005*; *Kräusslich et al., 1995*; *Liang et al., 2002*; *Liang et al., 2003*; *Melamed et al., 2004*; *Pettit et al., 1994*; *Rihn et al., 2013*; *von Schwedler et al., 2003*). Low resolution electron microscopy studies indicate that the junction forms what appears to be a pillar of density or helical bundle, and so is likely to stabilize the Gag lattice (*Schur et al., 2015a*; *Schur et al., 2015b*; *Wright et al., 2007*). However, neither the SP1 spacer nor the putative helical bundle is a universal feature of retroviral Gag shells (*Bharat et al., 2012*). This suggests that the SP1 spacer is not generally important for Gag assembly, or that perhaps the CA-SP1 junction has another, as yet structurally undefined assembly determinant.

Structural studies of the HIV-1 CA-SP1 junction in context of longer Gag fragments (CTD-SP1 and CTD-SP1-NC) have shown that the junction is primarily disordered (*Newman et al., 2004*; *Worthylake et al., 1999*). Nevertheless, the backbone NMR chemical shifts of junction residues deviate from expected random coil values, indicating a small propensity towards an α-helical conformation (*Newman et al., 2004*). Indeed, the isolated SP1 peptide displays concentration dependent secondary structure in aqueous solution, and folds into an α-helix at the millimolar protein concentrations found in virions (*Datta et al., 2011*). Nucleic acids (both RNA and DNA) can promote Gag assembly in vitro, presumably because of proximity-induced interactions between multiple copies of Gag that bind to the same nucleic acid molecule (*Campbell and Rein, 1999*; *Campbell and Vogt, 1995*; *Gross et al., 2000*). It therefore seems that nucleic acid-induced Gag clustering might also promote folding of the SP1 helix.

To learn how the CA-SP1 junction functions as a molecular switch during HIV-1 assembly and maturation, we determined the structure of its immature, assembled form by X-ray crystallography at a resolution of 3.27 Å. Our analysis elucidates how local conformational changes in the CA-SP1 switch

promote Gag assembly, drive large-scale capsid rearrangements, and regulate proteolysis during maturation. We also gained new insights on the mechanism of action of maturation inhibitors.

## Results

### HIV-1 CA CTD-SP1 recapitulates the immature Gag lattice

In the immature HIV-1 Gag shell, the NTD, CTD, and SP1 regions form three layers of lattice-stabilizing interactions (*Schur et al., 2015b*; *Wright et al., 2007*). The CTD-SP1 fragment of HIV-1 Gag contains the minimal information required to assemble the immature lattice, whereas the NTD is dispensable (*Accola et al., 2000*). Accordingly, we found that purified CTD-SP1 protein assembled in vitro into flat sheets with the subunits organized into a hexagonal lattice with the expected unit cell spacing of the CTD layer of the immature Gag lattice (~74 Å) (*Wright et al., 2007* ) (*Figure 1—figure supplement 1*). We then performed crystallization screens using a large number of commercial and in-house precipitants. We obtained many crystal hits, but invariably these were of the mature-like dimer form of the CTD with disordered SP1 tails, as observed previously (*Worthylake et al., 1999*). Using electron diffraction, we identified small and scarce plate crystals that gave a hexagonal diffraction pattern and unit cell spacing close to 70 Å (not shown). Upon optimization, these crystals diffracted X-rays to about 3.27 Å resolution (mean I/σI $\geq$ 1), and we determined the crystal structure to $R/R_{free}$ values of 0.246/0.278 (*Table 1* and *Figure 1—figure supplement 2*). Our *post hoc* analysis is that these crystals were rare because the CA-SP1 junction residues had to fold and become ordered during crystallization. We speculate that this rate limiting step of crystallization reflects the behavior of the junction during assembly of HIV-1 Gag.

The CTD-SP1 plate crystals were made up of stacked sheets of flat hexagonal lattices, with each sheet consisting of CTD-SP1 hexamers connected by CTD dimer linkages (*Figure 1A*). Our structure therefore represents a flattened version of the immature HIV-1 Gag lattice. The entire CTD-SP1 hexamer – including both the main CTD fold and ordered SP1 residues – gave an excellent fit to an 8.8 Å resolution cryoEM map of the immature HIV-1 Gag lattice (*Schur et al., 2015b*) (*Figure 1—figure supplement 3A*). The dimer also gave a good fit (*Figure 1—figure supplement 3B*), but in this case the SP1 helix was more laterally displaced from its corresponding density in the cryoEM map (asterisk in *Figure 1—figure supplement 3B*). This is likely because the cryoEM structure is of a curved Gag lattice, whereas our crystal structure is of a flat lattice. To gain insight on how loss of curvature in our crystal lattice is accommodated by the subunits, we superimposed the crystal structure with the model reported by Schur et al., which was created by flexible fitting of the main CTD fold (but not the CA-SP1 junction) into the cryoEM map (PDB 4USN) (*Schur et al., 2015b*). We found that superposition of the hexamer units resulted in an average root mean square displacement of 1.92 Å over equivalent Cα atoms, whereas superposition of the dimers resulted in a comparable displacement of 1.93 Å. We therefore surmise that any changes in tertiary or quaternary structure that were caused by flattening of the lattice in our crystals have been dispersed across the subunits. Unlike the mature capsid, which contains sharp pentameric declinations and a variably curving lattice (*Ganser et al., 1999*; *Pornillos et al., 2011*; *Zhao et al., 2013*), the immature HIV-1 Gag shell is a more smoothly curving sphere that is interrupted by large discontinuities (*Briggs et al., 2009*; *Keller et al., 2011*; *Wright et al., 2007*). It therefore seems that generation of Gag curvature does not necessarily require the kinds of conformational variations observed in the mature capsid subunits.

### The CA-SP1 junction is structurally bipartite

In our crystal structure, each subunit contains the main CTD fold (amino acids 281–351 in the HIV-1 Gag numbering scheme) and the CA-SP1 switch region (residues 352–370). The CTD adopts the canonical fold: a short $3_{10}$ helix, followed by a strand/turn element called the major homology region (MHR), and then four α-helices (helices 8–11) (*Gamble et al., 1997*; *Worthylake et al., 1999*) (*Figure 2A*). The switch region immediately follows the main CTD fold, and consists of two parts: a type II β-turn (residues 352–355) and an α-helix that spans the CA-SP1 junction (residues 356–370). This bipartite character was not anticipated but could explain the apparent structural variability in the C-terminal switch regions of immature retroviral capsids, which were discerned from subnanometer resolution cryoEM maps (*Bharat et al., 2012*; *Schur et al., 2015a*; *2015b*).

**Table 1.** Structure statistics for HIV-1 Gag CTD-SP1.

| Diffraction | |
| --- | --- |
| Beamline | APS 22ID |
| Wavelength (Å) | 1.0 |
| Processing program | HKL2000 |
| Space group | C2 |
| Cell dimensions | $a$ = 70.96 Å |
| | $b$ = 122.73 Å |
| | $c$ = 85.41 Å |
| | $\alpha = \gamma = 90°$, $\beta = 94.3°$ |
| Resolution range, Å | 50-3.27 (3.42-3.27) |
| $R_{merge}$ / $R_{pim}$ | 0.22 (0.74) / 0.11 (0.47) |
| Mean I/σ<I> | 5.99 (1.28) |
| Completeness,% | 87.0 (66.4) |
| Average redundancy | 3.7 (2.5) |
| Wilson B-factor, Å$^2$ | 85.21 |
| **Refinement** | |
| Refinement program | PHENIX |
| Resolution range | 42.59-3.27 (3.45-3.27) |
| No. of unique reflections | 9,710 (908) |
| Reflections in free set | 1,009 (88) |
| $R_{work}$ | 0.246 (0.369) |
| $R_{free}$ | 0.278 (0.408) |
| **No. of nonhydrogen atoms** | |
| protein | 3,865 |
| solvent | 0 |
| *Average B-factor, Å$^2$* | |
| protein | 84.09 |
| solvent | n/a |
| Coordinate deviations | |
| bond lengths, Å | 0.003 |
| bond angles, ° | 0.513 |
| **Validation and Deposition** | |
| *Ramachandran plot* | |
| favored,% | 98.9 |
| outliers,% | 0 |
| MolProbity clash score | 0.13 |
| PDB ID | 5I4T |

Values in parenthesis are for the highest resolution shell.

Comparison of the immature CTD-SP1 subunit with the mature CTD subunit reveals a local conformational change (presence or absence of a kink) in the dimerization helix, as noted previously (*Bartonova et al., 2008*) (arrows in *Figure 2A,B*). This correlates with the different orientations of the subunits across the immature and mature dimer interfaces (*Figure 2—figure supplement 1*) (*Bartonova et al., 2008*; *Gres et al., 2015*). Another difference is that, in the mature CTD, residues downstream of helix 11 are disordered (*Bartonova et al., 2008*; *Gamble et al., 1997*;

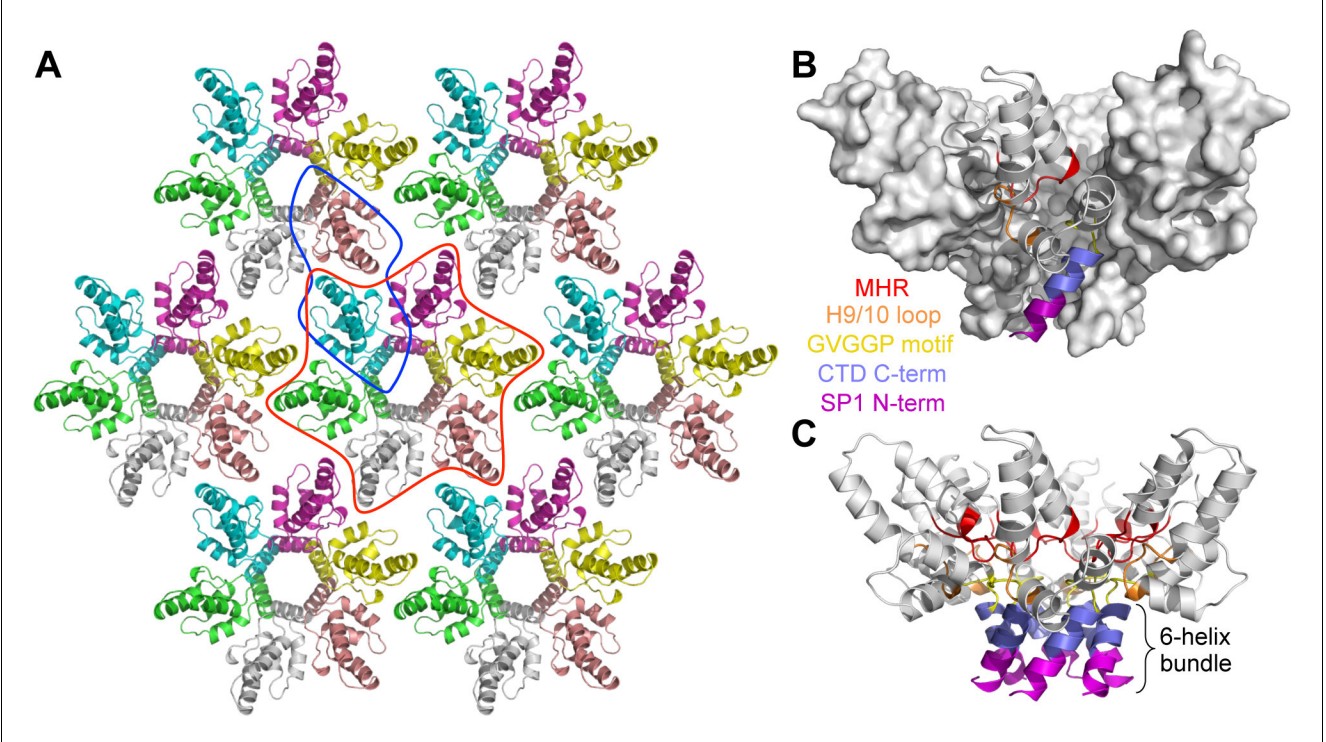

**Figure 1.** Crystal structure of the immature HIV-1 Gag CTD-SP1 lattice. (**A**) Top view of the lattice, with symmetry equivalent subunits in the same color. A hexameric unit is outlined in red, and a dimeric unit is outlined in blue. (**B,C**) Side views of the hexamer. The structural elements that make intermolecular contacts are colored and labeled: MHR (red), helix 9/10 loop (orange), GVGGP β-turn motif (yellow), and junction helix (blue/magenta).

The following figure supplements are available for figure 1:

**Figure supplement 1.** Initial characterization of CTD-SP1.

**Figure supplement 2.** Maps and model building.

**Figure supplement 3.** Comparison with the 8.8 Å cryoEM map of the immature HIV-1 Gag hexamer.

*Worthylake et al., 1999*) (*Figure 2B,C*). These observations provide further support for the presumption that the C-terminal tail of CA becomes unfolded upon maturation and has no apparent role in assembly of the mature capsid.

The CTD-SP1 hexamer looks like a goblet, with the main CTD fold forming the cup and the CA-SP1 junction forming the stem (*Figure 1B,C*). Lateral packing between the subunits is quite loose along the body of the cup, and close contacts involving the main CTD fold occur only where the cup meets the stem. In this region, the MHR loops line the bottom inner surface of the cup (*Figure 1B,C*, colored in red), whereas the loops connecting helices 9 and 10 (9/10 loop, orange) line the outer surface of the cup and contact the tops of the junction helices in the stem (blue/magenta). Underneath the cup, the junction helices form a 6-helix bundle as predicted (*Accola et al., 1998*; *Wright et al., 2007*). All of the aliphatic sidechains in the junction helix make 'knobs-in-holes' interactions similar to classical coiled-coils (*Figure 3A–C*).

## The CA-SP1 junction helix regulates proteolysis by sequestering the scissile bond

During HIV-1 maturation, the CA-SP1 junction is one of ten cleavage sites in the precursor Gag and Gag-Pol polyproteins that are cleaved by the viral protease (PR). Proteolytic processing occurs at different rates, with the CTD-SP1 junction processed with the slowest rate, and the downstream SP1-NC junction the fastest (reviewed in *Lee et al., 2012*). The C-terminal boundary of the junction helix

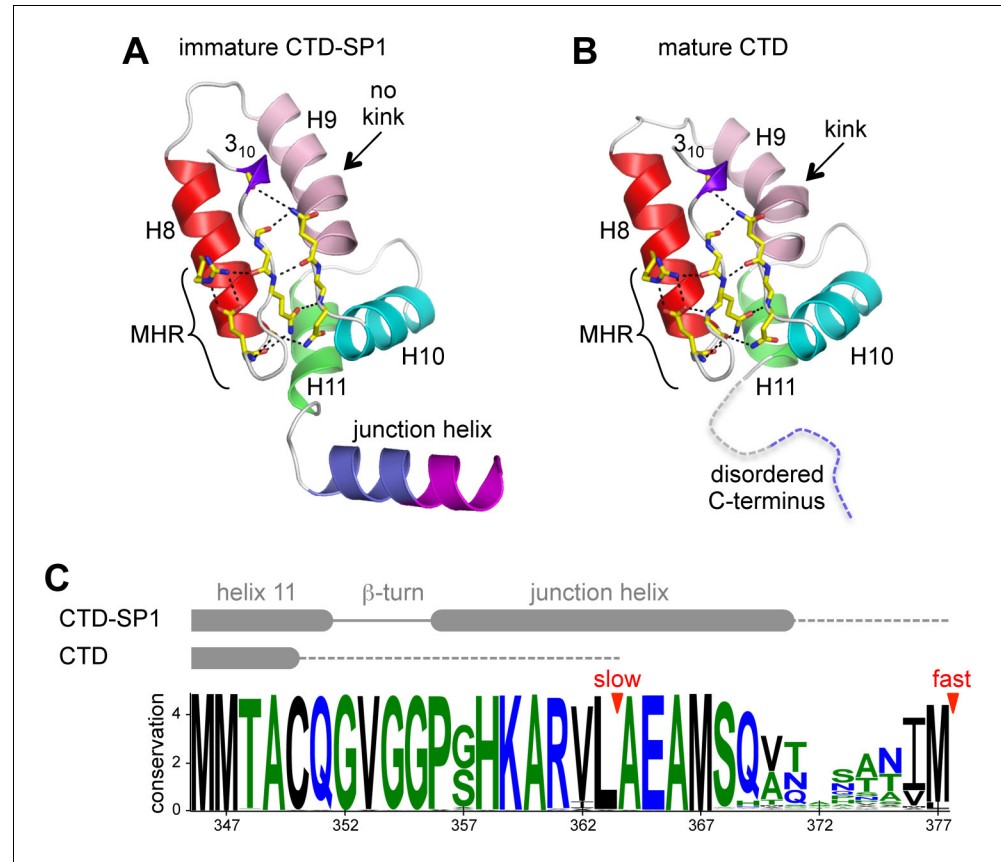

**Figure 2.** Comparison of immature and mature forms of the CTD-SP1 switch. (**A**) The immature CTD-SP1 subunit. Helices are shown as colored ribbons. Absolutely conserved MHR residues that mediate a salt bridge/hydrogen bond network are shown as yellow sticks, as are contributing residues from the helix 9/10 loop. (**B**) Equivalent view of the mature CTD (PDB 1A43) (***Worthylake et al., 1999***). Disordered residues at the C-terminus are in dashes. A pronounced kink in the dimerization helix (H9, pink) is absent in the immature form (black arrows). (**C**) Sequence conservation in the CTD-SP1 junction, derived from the curated Los Alamos HIV sequence database (***Kuiken et al., 2003***) (6824 sequences). Secondary structure of the immature and mature forms are shown above, with dashes indicating disordered regions. Proteolytic processing sites are marked by red arrowheads.

The following figure supplement is available for figure 2:

**Figure supplement 1.** Quasi-equivalent conformations of the immature and mature dimers.

tracks precisely with sequence conservation; downstream residues that are disordered in our crystal structure are not conserved (***Figure 2C***). This is consistent with the notion that the downstream SP1-NC junction has the fastest rate of processing because it is disordered in the immature virion.

In our structure, the scissile bond (between Leu363 and Ala364) is located inside the helical barrel (yellow spheres in ***Figure 3A***), where it is sequestered within a pocket covered by a methionine side-chain (Met367) (***Figure 3D***). When the CA-SP1 junction binds to PR, it adopts a fully extended, β-strand configuration wherein sidechains on either side of the scissile bond occupy sub-pockets within the enzyme active site (***Prabu-Jeyabalan et al., 2000***) (***Figure 3E***, right). In the CTD-SP1 hexamer, these same sidechains mediate the 'knobs-in-holes' interactions of the 6-helix bundle (***Figure 3E***, left). Therefore, for PR to gain access to its substrate, the 6-helix bundle must unfold. Since proteolysis rates for the different cleavage sites are comparable when measured with peptide substrates (***Lee et al., 2012***), the rate limiting step for the CA-SP1 site is unfolding of the 6-helix bundle. This explains why the CA-SP1 junction displays the slowest rate of proteolysis and is the final trigger that elicits HIV-1 maturation.

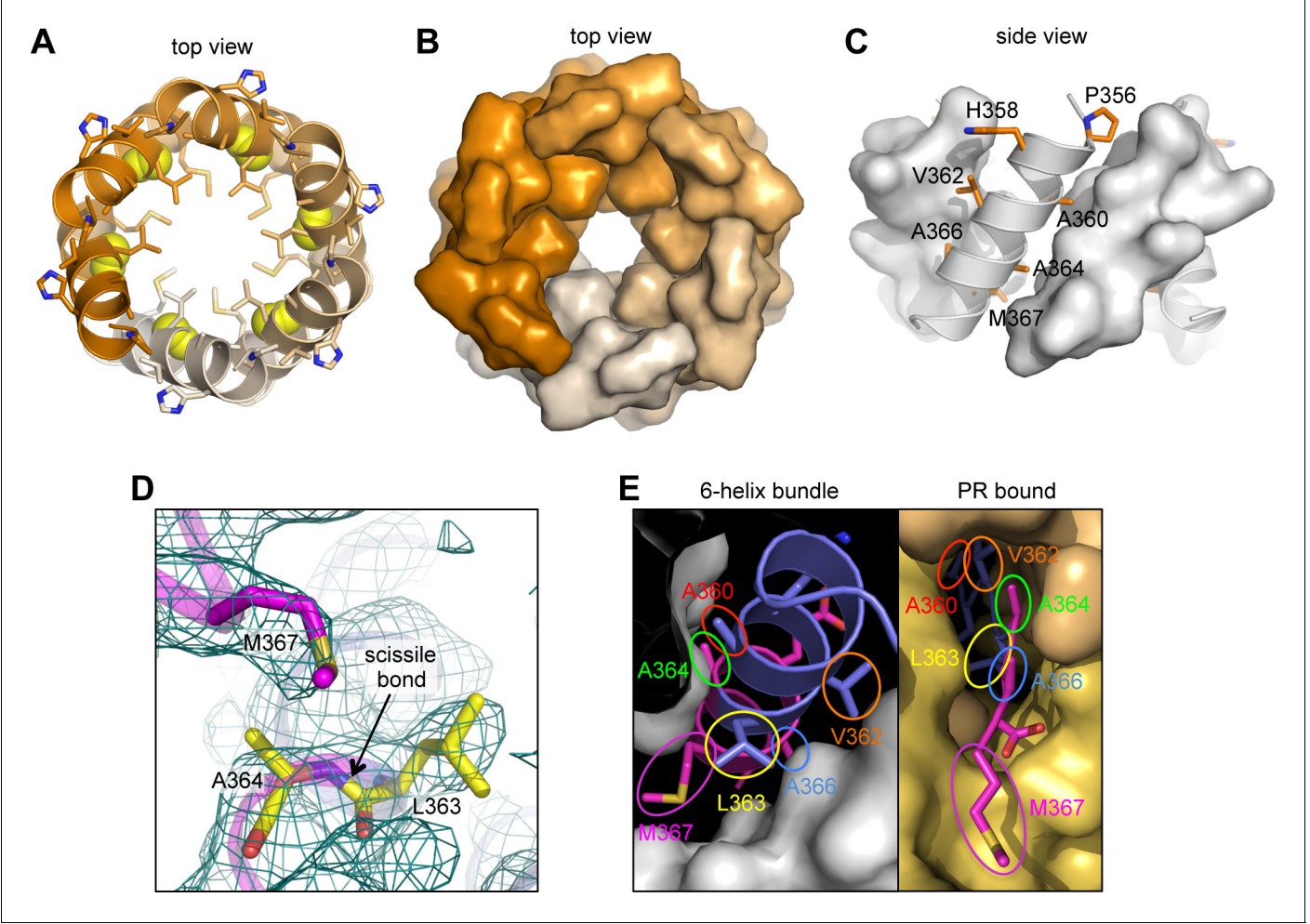

**Figure 3.** The junction 6-helix bundle. (A) Top view of the 6-helix bundle. The helical backbone is in ribbons, and sidechains that mediate 'knobs-in-holes' type packing are shown as sticks. Yellow spheres indicate the scissile peptide bond between Leu363 and Ala364 (CA residue Leu231 and SP1 residue Ala1). (B) Same top view, with the subunits rendered in surface representation. (C) Side view, with alternating subunits rendered as ribbons or surfaces. 'Knobs-in-holes' sidechains are shown as orange sticks and labeled. (D) Close-up of the scissile bond, which is sequestered at the bottom of a pocket occupied by the Met367 sidechain. Mesh shows a composite simulated annealing omit map (1σ). (E) Comparison of the CA-SP1 junction in context of the 6-helix bundle (left) and bound to the viral protease active site (right) (PDB 1 KJH) (*Prabu-Jeyabalan et al., 2000*). Sidechains that mediate both types of interactions are encircled, with equivalent residues in the same color.

## The β-turn organizes multiple hexamer-stabilizing elements

Our structure revealed a second structural component of the CA-SP1 switch, which consists of five amino acid residues that connect the main fold of the CTD and the junction helix ($_{352}$GVGGP$_{356}$) (*Figure 4A*). This motif makes a right-handed type II β-turn that buries the Val353 sidechain within a shallow sub-pocket at the bottom of the CTD, which is made of the MHR and 9/10 loops. In context of the hexamer, the entire β-turn is almost completely buried within a larger pocket made up of two MHR loops, two 9/10 loops, and the junction helices from three adjacent subunits (*Figure 4B,C*). Thus, the β-turn brings together multiple structural elements to generate the Gag hexamer. This key structural role indicates that the β-turn is a critical Gag assembly determinant.

## The entire CA-SP1 junction is required for immature Gag assembly in vitro

Numerous mutagenesis studies have established the importance of the CA-SP1 switch in assembly of the HIV-1 Gag protein (*Accola et al., 1998*; *Datta et al., 2011*; *Guo et al., 2005*;

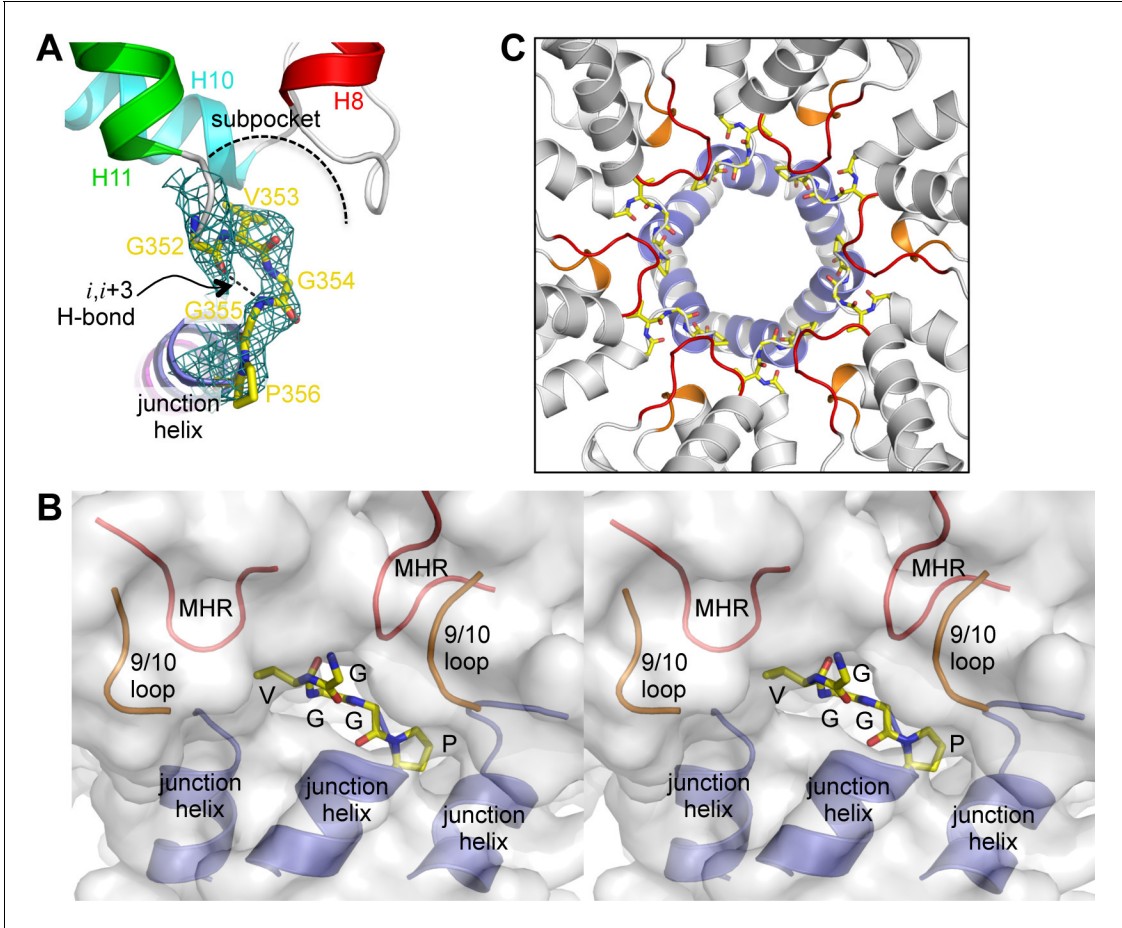

**Figure 4.** The β-turn 'clasp' motif. (**A**) Close-up view of the GVGGP motif in context of a single subunit. An *i,i+3* hydrogen bond characteristic of a right-handed β-turn is indicated, as is the sub-pocket occupied by the Val353 sidechain. Mesh shows a composite simulated annealing omit map (1σ). (**B**) Stereo view of the β-turn and its surrounding pocket, which is made up of two MHR loops (red), two helix 9/10 loops (orange), and the N-terminal ends of three junction helices (blue). The hexamer is rendered as a translucent surface. (**C**) The six β-turns (yellow sticks) in context of the hexamer. The surrounding MHR, 9/10 loops, and junction helices are colored as in (**B**).

The following figure supplement is available for figure 4:

**Figure supplement 1.** Comparison of the HIV and MPMV switch regions.

*Kräusslich et al., 1995*; *Liang et al., 2002*; *Liang et al., 2003*; *Melamed et al., 2004*; *Pettit et al., 1994*). Assembly defects in previous studies were assessed in different ways, however, so we performed our own alanine scan to obtain a uniform dataset. Mutations were made in context of the previously described △MA-CA-SP1-NC HIV-1 Gag construct, which displays pH-dependent assembly behavior upon dialysis with a single-stranded DNA template (*Gross et al., 2000*). At high pH, △MA-CA-SP1-NC assembles into thick-walled immature virus-like particles, whereas at low pH, it assembles mature-like tubes, cones, and spheres. The immature △MA-CA-SP1-NC VLPs are nearly identical to the immature Gag shell of authentic virions (*Briggs et al., 2009*; *Gross et al., 2000*).

When we dialyzed 2 mg/mL of wildtype (WT) △MA-CA-SP1-NC with a 26-mer DNA oligonucleotide at pH 8, we obtained immature VLPs as expected, with little evidence of off-pathway assemblies or aggregation (*Figure 5B* and *Figure 5—figure supplement 1*). Under the same conditions, we obtained three mutant phenotypes (which are color coded in *Figure 5A* and summarized in *Table 2*; a representative image of each mutant is shown in *Figure 5—figure supplement 1*): WT-like mutants assembled immature VLPs (green, +) (*Figure 5B*), 'non-assembling' mutants primarily produced aberrant particles or aggregates (magenta, −) (*Figure 5C*), and intermediate mutants formed

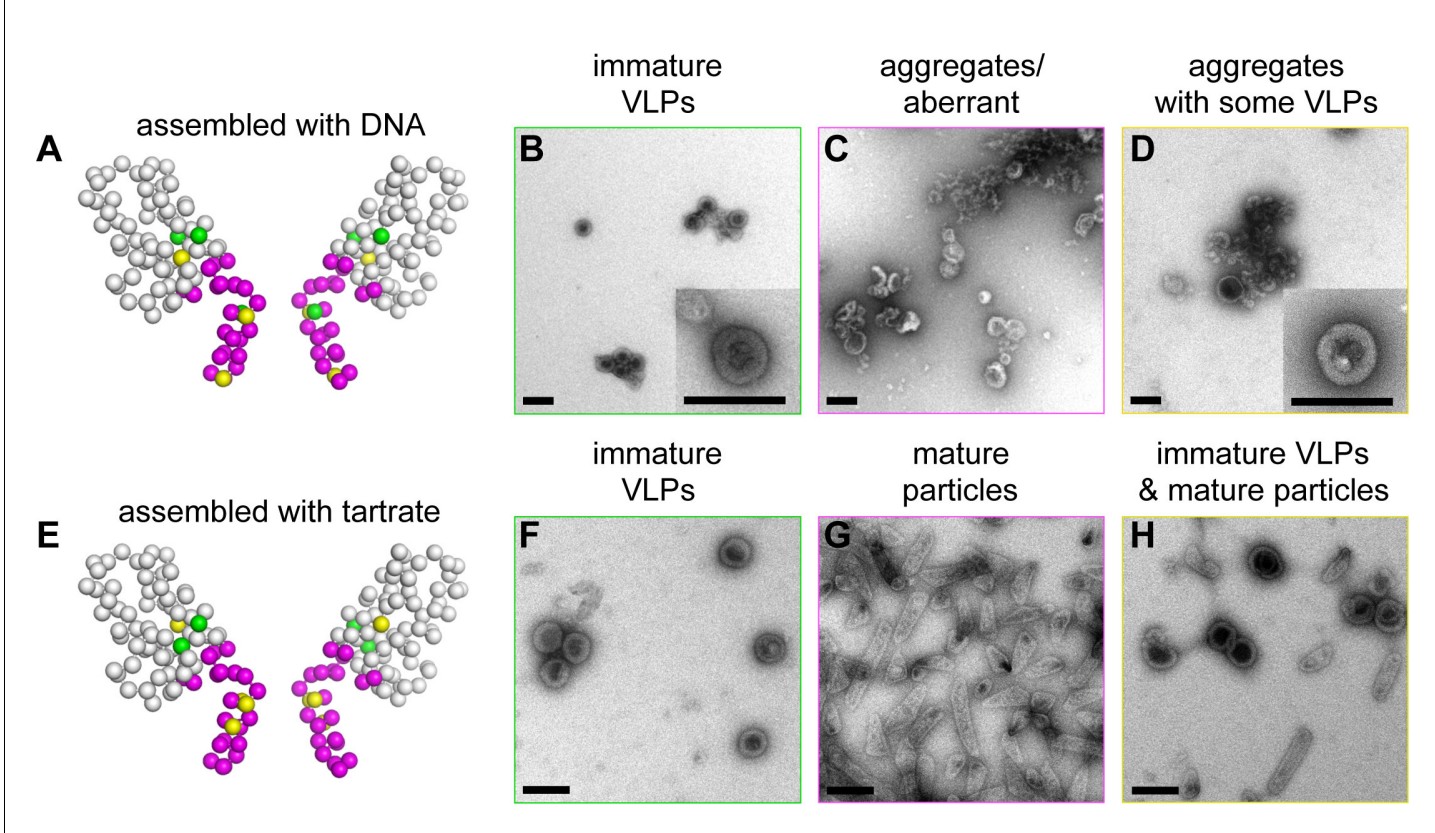

**Figure 5.** Summary of structure-based alanine scanning mutagenesis. (A–D) Phenotypes of HIV-1 Gag △MA-CA-SP1-NC assembled with DNA. The ribbon diagram is of two opposing subunits in the hexamer with Cα atoms shown as spheres and color-coded according to assembly phenotypes as shown in panels B, C, and D. (E–H) Phenotypes of △MA-CA-SP1-NC assembled with tartrate. Scale bars = 150 nm.

The following figure supplements are available for figure 5:

**Figure supplement 1.** Alanine scanning mutagenesis of CA-SP1.
**Figure supplement 2.** Alanine scanning mutagenesis of CA-SP1.

both immature VLPs and aberrant particles/aggregates (yellow, +/–) (*Figure 5D*). We also observed mature-like tubes for some of the mutants, but these were relatively rare (indicated by asterisks in *Table 2*; representative images in *Figure 5—figure supplement 1*).

As expected from the structure, mutations in the main fold of the CTD that are located within the loosely packed cup region of the hexamer (R286A and R294A) were WT-like, whereas CTD mutations closer to the tightly packed junction region (M347A and Q351A) were more disruptive of assembly (*Table 2* and *Figure 5—figure supplement 1*). In contrast, virtually all of the 19 single-point mutations in the structured region of the CA-SP1 switch (residues 352–370) resulted in aberrantly assembled particles and/or protein aggregates (*Figure 5A*, *Table 2*, and *Figure 5—figure supplement 1*). Two exceptions were G357A and K359A, both of which are located near the N-terminal end of the junction helix. The G357A phenotype was not unexpected because this is a polymorphic position (*Figure 2C*). K359A, on the other hand, replaced a lysine residue lining the inside of the helical barrel with a hydrophobic residue, and the substitution may have eliminated clashes between like charges inside the 6-helix bundle. Thick-walled immature VLPs were also obtained with the S368A mutation near the C-terminal end of the junction helix.

Apart from residues close to the C-terminal end of the junction helix, many of the switch mutants appeared to behave in an 'all-or-none' manner (i.e., they either assembled immature VLPs or aberrantly), indicating that both the β-turn and junction helix must be functional for assembly to occur in

**Table 2.** In vitro assembly phenotypes of HIV-1 △MA-CA-SP1-NC Gag.

| Mutation | Location | DNA template[1] | Tartrate[2] |
|---|---|---|---|
| Wildtype | | + | + |
| R286A | MHR strand | + | +/m |
| P289A | MHR loop | – | m |
| K290A | MHR loop | – | m |
| R294A | MHR helix | + | + |
| P328A | 9/10 loop | – | m |
| D329A | 9/10 loop | – | m |
| M347A | Helix 11 | +/– | + |
| Q351A | Helix 11 | – | m |
| G352A | β-turn | – | m |
| V353A | β-turn | – | m |
| G354A | β-turn | – | m |
| G355A | β-turn | – | m |
| P356A | β-turn /Junction helix | – | m |
| G357A | Junction helix | +/– | +/m |
| H358A | Junction helix | – | m |
| K359A | Junction helix | + | +/m |
| A360V | Junction helix | – | m |
| R361A | Junction helix | – | +/m |
| V362A | Junction helix | – | m |
| L363A | Junction helix | – | m |
| A364V | Junction helix | – | m |
| E365A | Junction helix | –* | m |
| A366V | Junction helix | – | m |
| M367A | Junction helix | – | m |
| S368A | Junction helix | +/– | m |
| Q369A | Junction helix | – | m |
| V370A | Junction helix | –* | m |
| T371A | Disordered | +/–* | +/m |
| N372A | Disordered | +/– | +/m |
| P373A | Disordered | +/–* | m |
| A374V | Disordered | +/– | +/m |
| T375A | Disordered | +/– | +/m |
| I376A | Disordered | +/– | +/m |
| M377A | Disordered | +/– | +/m |

[1] +, as in *Figure 5B*; –, as in *Figure 5C*; +/–, as in *Figure 5D*

[2] +, as in *Figure 5F*; m, as in *Figure 5G*; +/m, as in *Figure 5H*

* rare mature tubes

vitro. Such a behavior was unexpected because in high-order assembly systems, local defects in protein-protein interactions are typically buffered by cooperativity. Therefore, the 'all-or-none' behavior must be due to a defect in nucleation. To test this idea more rigorously, we used tartrate salts to induce assembly of the △MA-CA-SP1-NC VLPs (*Figure 5E*). The absence of nucleic acids ensured that nucleation was triggered only by protein-protein interactions. In this format, most of the CA-

SP1 switch mutants again failed to form immature VLPs, but now instead assembled efficiently into thin-walled mature-like particles (*Figure 5E–H*, *Table 2*, and *Figure 5—figure supplement 2*). The same phenotype is observed when the entire SP1 spacer is deleted (*Gross et al., 2000*). Since junction residues downstream of the main CTD fold are not required for assembly of the mature lattice and are almost certainly unfolded in the thin-walled particles, these results further indicate that nucleation of the immature particles requires folding of both the β-turn and junction helix.

Some of the mutants formed both immature VLPs and mature particles in the presence of tartrate, in similar proportions (R286A, G357A, K359A, R361A) (*Figure 5H* and *Figure 5—figure supplement 2*). Examination of well-separated particles indicated that each particle was either immature-like or mature-like, i.e., we did not observe ones that were partly thin-walled and partly thick-walled. We believe that this is an important observation, because it indicates that the identity of the particle is established at nucleation (*Gross et al., 2000*), and is more supportive of a disassembly/reassembly pathway of HIV-1 capsid maturation than a displacive or condensation mechanism (*Keller et al., 2013*). Another notable mutant was K359A, which assembled into immature VLPs in the presence of DNA (*Figure 5—figure supplement 1*), but assembled into predominantly mature-like particles by the tartrate method (*Figure 5—figure supplement 2*). These results indicate that, although the HIV-1 Gag protein does not require nucleic acid to assemble into immature particles in vitro, folding of the CA-SP1 switch can be promoted by nucleic acid, likely due to clustering.

We then extended our alanine scan to include SP1 residues downstream of the junction helix (residues 371–377). As expected, all of these mutants were competent to form immature VLPs in the presence of DNA, but surprisingly, were still significantly impaired in assembly efficiency (*Table 2* and *Figure 5—figure supplement 1*). By the tartrate method, the mutants formed both mature and immature VLPs, with the exception of P373A, which was predominantly mature (*Table 2* and *Figure 5— figure supplement 2*). Our interpretation of these results is that, even though the C-terminal end of SP1 is not highly conserved in sequence (*Figure 2C*), the mutants were still defective in folding of the CA-SP1 junction, and again, nucleic acid (DNA) can partially compensate for the folding defect. Alternatively, it is possible that the helical character of SP1 extends downstream all the way to the NC domain, as previously proposed (*Morellet et al., 2005*). However, our crystal structure, the cryoEM reconstruction (*Figure 1 —figure supplement 3*) (*Schur et al., 2015a*), and sequence conservation (*Figure 2C*) appear to be in good agreement as to the boundaries of the junction helix. We therefore conclude that our structure defines the core helical bundle, although an extended helix may occur in a fraction of Gag molecules, for example the ones that nucleate assembly.

Finally, we tested mutations at the MHR loop (P289A, K290A) and the 9/10 loop (P328A, D329A) that surround the Val353 pocket at the bottom of the CTD. These mutants behaved in a similar way as the β-turn and junction helix mutations (*Figure 5—figure supplement 1*, *Figure 5— figure supplement 2*, and *Table 2*). These results confirm that binding of Val353 to the bottom of the CTD – and by inference, proper folding of the β-turn – is a critical step in nucleation of the immature VLPs. We speculate that the valine pocket at the bottom of the CTD functions as a folding chamber or 'chaperone' for the β-turn.

## Discussion

Our crystal structure of CTD-SP1 represents the highest resolution structure of the immature HIV-1 Gag lattice to date, and confirms prediction that the CA-SP1 junction forms a 6-helix bundle to sequester the PR cleavage site. The CA-SP1 junction elicits the slowest rate of proteolysis during maturation because the 6-helix bundle must unfold to bind PR. Binding energy might be sufficient to induce unfolding, but this may be unlikely since PR has low intrinsic affinity for its substrates (mM range in vitro (*Pettit et al., 1994*)). Another attractive possibility is that PR gains substrate access by transient, cooperative unfolding and refolding of the 6-helix bundle (*Bush and Vogt, 2014*). Our proposed mechanism is analogous to the predicted transient unfolding of the collagen triple helix that permits cleavage by matrix metalloproteases (*Lu and Stultz, 2013*). In non-enveloped viruses, such a mechanism allows surface exposure of otherwise internal and structured capsid protein segments, which then permits access to antibodies and exogenously added proteases (*Bothner et al., 1998*; *Lin et al., 2012*).

Our analysis also supports a model wherein maturation inhibitors like bevirimat deny PR access, not by sequestering the scissile bond (the 6-helix bundle already does this), but rather by stabilizing the 6-helix bundle and inhibiting its unfolding. This model predicts that resistance can develop by means of mutations that destabilize the junction. Such mutations would enhance proteolysis in the absence of the maturation inhibitor, and indeed this seems to be a general property of well-characterized escape mutations (*Adamson et al., 2006*; *Li et al., 2006*; *Sakalian et al., 2006*; *Zhou et al., 2004*). In fact, some escape mutations that have been selected for in tissue culture appear to destabilize the Gag hexamer altogether, and the mutant viruses become dependent on the inhibitor to assemble (*Waki et al., 2012*). Conversely, mutations that stabilize the 6-helix bundle would be expected to inhibit proteolysis even in the absence of inhibitor. One such mutation is T371I (*Fontana et al., 2015*). This residue immediately follows the junction helix, and we speculate that the isoleucine substitution stabilizes and extends the helix. Extending the helix will place this side-chain inside the barrel underneath a ring of methionine residues, where it can provide additional non-native hydrophobic contacts. We have not yet been able to determine whether bevirimat binds outside or inside the bundle, but the compound has been crosslinked to both the junction helix and the MHR (*Nguyen et al., 2011*), which seems consistent with the second possibility.

Another important finding in this study is that the HIV-1 CA-SP1 switch has a second structural component: a β-turn that connects the junction helix to the main CTD fold. The β-turn is almost completely buried within an assembly-induced pocket surrounded by three adjacent subunits. Both its location and the fact that it interacts with 7 other hexamer-stabilizing structural elements indicate that the β-turn is a critical Gag assembly determinant, and may act as a 'clasp' that locks the interactions.

Although our in vitro assembly experiments indicated that virtually the entire CA-SP1 switch region is intolerant of mutations, we did observe that assembly defects were graded – the most severe mutations mapped to the β-turn and severity fell off towards the C-terminus of the junction helix (i.e., the SP1 spacer). A similar trend has been observed in previous studies, and indeed, the gradation in assembly defect is significantly more pronounced in cells (*Accola et al., 1998*; *Al-Mawsawi et al., 2014*; *Datta et al., 2011*; *Guo et al., 2005*; *Kräusslich et al., 1995*; *Liang et al., 2002*; *Liang et al., 2003*; *Melamed et al., 2004*; *Rihn et al., 2013*; *von Schwedler et al., 2003*). For example, HIV-1 Gag that completely lacks the SP1 spacer can still assemble into large, membrane-associated patches (*Kräusslich et al., 1995*), whereas β-turn mutants do not form plasma membrane-associated patches and remain diffusely distributed throughout the cytosol (*Liang et al., 2003*). These observations indicate that SP1 is not strictly required for the fundamental higher-order self-association property of HIV-1 Gag. On the other hand, the β-turn is required very early in assembly, perhaps during nucleation of the viral particle.

In context of full-length HIV-1 Gag, the entire CA-SP1 switch region was suggested to be unstructured but with weak helical propensity (*Newman et al., 2004*; *Worthylake et al., 1999*). A fully folded β-turn (but not junction helix) was also observed in the crystal structure of the immature-like dimer form of the CTD (*Bartonova et al., 2008*). We therefore speculate that in the unassembled, soluble Gag protein, the β-turn fluctuates between folded and unfolded states, and that the entire CA-SP1 junction folds only upon assembly of the immature lattice. Since we found that an intact β-turn sequence cannot rescue a defective junction helix and vice versa (at least in vitro), we propose that folding of the CA-SP1 switch, and in particular the β-turn, is a critical step in nucleation of the immature Gag lattice.

Assembly of retroviral particles is thought to initiate when the viral RNA genome dimerizes and exposes high-affinity binding sites for the Gag C-terminal NC domain (*D'Souza and Summers, 2004*; *Keane et al., 2015*). The resulting ribonucleoprotein complex, containing several Gag molecules, is reasonably assumed to nucleate assembly of the immature viral particle. RNA-mediated Gag clustering has been proposed to nucleate assembly by promoting the CTD dimerization interaction (*Amarasinghe et al., 2000*; *Bush and Vogt, 2014*; *Ma and Vogt, 2004*). Since the isolated SP1 peptide folds into a helix in a concentration dependent manner (*Datta et al., 2011*), we suggest instead that RNA-mediated clustering directly promotes folding of the CA-SP1 junction and the 6-helix bundle and, thereby, formation of Gag hexamers. Compared to a dimer, a hexamer is more intuitively appealing as the nucleator of the spherical Gag shell, since it is the smallest unit comprising the set of protein-protein interactions that are exclusive to the immature lattice and not found in the mature capsid or other functional states of the protein. We further speculate that folding of the

junction will have the additional role of reeling in the NC domain and its bound genome, increasing the probability that the accumulating Gag molecules will bind to the same RNA. It is likely that the RNA, in turn, exerts some corresponding pulling force on the assembling Gag lattice, helping to promote its encapsidation. Our analysis therefore supports the proposal that the CA-SP1 junction allosterically communicates the status of downstream NC/RNA events to the upstream assembly domains of Gag (*Bush and Vogt, 2014*).

In summary, our studies reveal the molecular details of a bipartite HIV-1 CA-SP1 switch with critical functions in both virus particle assembly and proteolytic maturation. Retroviruses that do not contain a spacer between their CA and NC domains might have eschewed sophisticated protease regulation but have retained the more fundamental assembly determinant at the C-terminal end of CA. An example is Mason-Pfizer monkey virus (MPMV), whose immature capsid structure does not contain a 6-helix bundle (*Bharat et al., 2012*). Sequence comparison confirms that MPMV does not have a spacer element, but does appear to have a 'clasp' motif that is similar to that of HIV-1 (*Figure 4—figure supplement 1*). Rous sarcoma virus (RSV) appears to be a contrasting example – it has a well-characterized spacer (*Bush et al., 2014*; *Keller et al., 2008*; *Schur et al., 2015a*), but does not have a recognizable turn motif. The RSV junction helix is longer and more stable than that of HIV-1 (*Bush et al., 2014*; *Keller et al., 2008*), and we surmise that the RSV spacer confers both the assembly and regulatory functions that are mediated by two distinct parts in HIV.

### Note added in proof

In a concurrent publication, Schur et al. report a cryoEM structure of the immature HIV-1 Gag lattice at near-atomic resolution (*Schur et al., 2016*). They also conclude that the CA-SP1 boundary folds into a 6-helix bundle that is important for assembly of Gag and its proteolytic maturation.

## Materials and methods

### Protein expression and purification

CTD-SP1 (residues 281–377 of HIV-1 NL4-3 Gag) was cloned with a non-cleavable His-tag (His$_6$-Gly$_2$) into pET30a (Novagen/EMD Millipore, Germany). The construct also contained the P373T substitution, which is a natural sequence variant (*Kuiken et al., 2003*). Protein was expressed in *E. coli* BL21 (DE3) cells by induction with 1 mM IPTG for 4 hr at 25°C in shake cultures. Bacteria were harvested by centrifugation and resuspended in 50 mM Tris, pH 8.3, 1 M LiCl, 10 mM β-mercaptoethanol (βME) supplemented with 0.3% (w/v) deoxycholate and protease inhibitor tablets (Roche). Cells were lysed by incubation with lysozyme and sonication. Lysates were clarified by centrifugation and then incubated with Ni-agarose resin (Qiagen, Germany) for 30 min at 4°C. Bound fractions were washed and eluted with a step gradient of 15–300 mM imidazole. The protein was purified to homogeneity using anion exchange and size exclusion chromatography in 20 mM Tris, pH 8.0, 0.5 M NaCl, 20 mM βME. Pure proteins were concentrated to 15–20 mg/mL.

### Two-dimensional crystallography

Screening for 2D crystals was performed as described (*Yeager et al., 2013*). CTD-SP1 (1 mM) was mixed with an equal volume of 0.4 M sodium-potassium tartrate and incubated overnight at room temperature. Samples were placed on a carbon-coated grid, washed with 0.1 M KCl, and preserved with 2% glucose in 0.1 M KCl. Low-dose images of vitrified samples were recorded with a Titan Krios transmission electron microscope (Philips/FEI, Hillsboro, OR) operating at 120 kV. A merged projection map (*Figure 1—figure supplement 1*) was calculated from 7 images, using the program 2dx (*Gipson et al., 2007*). A B-factor of -500 Å$^2$ was imposed to sharpen the map.

### X-ray crystallography

Screening for three-dimensional crystals was performed using a large number of commercial and in-house precipitants. Plate crystals that formed in 0.1 M Bis-Tris propane, pH 7–8, 0.8–1.0 M LiSO$_4$ were initially identified by electron diffraction as being composed of stacked hexagonal sheets. Crystals for X-ray diffraction experiments were optimized in sitting drops, which were set up at a 1:2 protein:precipitant ratio. We found that the best diffracting crystals formed when drops were made with freshly purified protein. Ethylene glycol (25%) in mother liquor was used as cryoprotectant.

Diffraction data were collected from a single crystal at beamline 22-ID at the Advanced Photon Source, and processed with HKL2000 (*Otwinowski and Minor, 1997*). The phase problem was solved by molecular replacement with an immature CTD hexamer model (PDB 4USN) (*Schur et al., 2015b*). Upon rigid body refinement, unbiased densities for the 6-helix bundle were readily observed in model-phased maps (*Figure 1—figure supplement 2A*). Multiple rounds of iterative model building and refinement were performed with the programs PHENIX (version 1.9–1692) (*Adams et al., 2010*) and Coot (*Emsley et al., 2010*). Due to the small size of the crystal (~20 microns in the longest dimension), the diffraction data were weak (mean $I/\sigma<I> = 6$ and completeness = 87%; *Table 1*). Nevertheless, we obtained very high quality maps for model building due to the fortuitous existence of 6-fold non-crystallographic symmetry (NCS), and through the use of modern density modification techniques implemented in PHENIX. To obtain the best unbiased map for building the CTD-SP1 junction, we first extensively refined the main CTD fold using reference model restraints (to PDB 3DS2) (*Bartonova et al., 2008*). A 6-fold NCS averaged map was then calculated, which clearly revealed helical densities (unbiased) for the junction (*Figure 1 — figure supplement 2B*). The junction helix was built into these densities as a polyalanine model using the 'Place Helix Here' command in Coot. After additional rounds of building and refinement, a feature-enhanced map was calculated with PHENIX (*Afonine et al., 2015*), which gave a unique solution to the helical registry (*Figure 1 —figure supplement 2C,D*). At low contour levels (~0.5 σ), residual densities that appeared to correspond to N-terminal His-tag residues were also observed, but these were left unmodeled. Secondary structure hydrogen bonding restraints, riding hydrogens, and local (torsion angle) 6-fold NCS restraints were used throughout the refinement process, as were structure validation tools implemented in both PHENIX and Coot. The current model was also validated with a composite simulated annealing omit map, shown in *Figure 3D*, *Figure 4A*, and *Figure 1—figure supplement 2E*. Structure statistics are summarized in *Table 1*.

## Alanine-scanning mutagenesis and in vitro assembly assays

For in vitro assembly assays, we used the △MA-CA-SP1-NC construct, which is a well-validated model system for the immature HIV-1 Gag shell (*Briggs et al., 2009*; *Gross et al., 2000*). WT and mutant proteins were expressed and purified as described (*Gross et al., 2000*). Assembly reactions were set up as follows. DNA templated assembly: Purified protein (148 μL at 2 mg/mL) was mixed with 26-mer single-stranded DNA oligonucleotide (5'-GGGAGTGGGGGGGACTGAAGCAATGAG-3') (2 μL at 1 mM) and dialyzed for 2 hr at 4°C into 1 L of 50 mM Tris, pH 8.0, 0.1 M NaCl, 1 mM EDTA, 2 mM βME. Particles were concentrated by centrifugation in a microcentrifuge at maximum speed for 10 min at 4°C, and resuspended in 15 μL of dialysis buffer. Tartrate-induced assembly: Protein (50 μL at 10–15 mg/mL) was mixed with 19.25 μL of 1.5 M tartrate and 7.7 μL of 1 M Tris, pH 7.5 and incubated for 2 hr at 37°C. This second approach was more efficient at promoting assembly, and so the centrifugation step was omitted.

For electron microscopy, each sample (3 μL) was applied to a glow-discharged, continuous carbon-coated grid for 2 min. Excess liquid was blotted off by touching the edge of the grid with filter paper. The grid was washed with distilled water, blotted, stained with 2% (w/v) uranyl acetate for 2 min, and blotted again. Images of the negatively stained samples were recorded using a Tecnai F20 transmission electron microscope (Philips/FEI) operating at 120 kV.

The assembly experiments were performed with two independent protein preparations for each mutant. In the second set of experiments, the grids were randomized so that the individual who acquired the images was unaware of the identity of the samples.

## Accession codes

Coordinates and structure factors are deposited in the Protein Data Bank under accession number 5I4T.

## Acknowledgements

We thank Dan Shi and Tamir Gonen for help and advice with electron diffraction experiments. This study was supported by NIH grants R01-GM066087 (MY, BKG-P, and OP), P50-GM082545 (MY), and P50-GM103297 (OP) JMW was supported by a postdoctoral NIH fellowship (F32-GM115007). X-ray diffraction data were collected at beamlines 22-BM and 22-ID at the Advanced Photon Source,

Argonne National Laboratory. Electron microscopy data were collected at the Molecular Electron Microscopy Core facility at University of Virginia. The Titan Krios microscope was funded in part by NIH grant S10-RR025067.

## Additional information

### Funding

| Funder | Grant reference number | Author |
|---|---|---|
| National Institutes of Health | R01-GM066087 | Mark Yeager<br>Barbie K Ganser-Pornillos<br>Owen Pornillos |
| National Institutes of Health | P50-GM082545 | Mark Yeager |
| National Institutes of Health | P50-GM103297 | Owen Pornillos |
| National Institutes of Health | F32-GM115007 | Jonathan M Wagner |

The funders had no role in study design, data collection and interpretation, or the decision to submit the work for publication.

### Author contributions

JMW, KKZ, JC, MDP, Acquisition of data, Analysis and interpretation of data, Drafting or revising the article; MY, BKG-P, OP, Conception and design, Acquisition of data, Analysis and interpretation of data, Drafting or revising the article

### Author ORCIDs

Owen Pornillos, http://orcid.org/0000-0001-9056-5002

## Additional files

### Major datasets

The following dataset was generated:

| Author(s) | Year | Dataset title | Dataset URL | Database, license, and accessibility information |
|---|---|---|---|---|
| Jonathan M Wagner, Kaneil K Zadrozny, Barbie K Ganser-Pornillos, Owen Pornillos | 2016 | Immature hexagonal lattice of HIV-1 Gag | http://www.rcsb.org/pdb/explore/explore.do?structureId=5I4T | Publicly available at the RCSB Protein Data Bank (accession no. 5I4T) |

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
