## [Decision Letter]

Thank you for submitting your article "Crystal structure of an HIV assembly and maturation switch" for consideration by *eLife*. Your article has been favorably evaluated by John Kuriyan (Senior editor) and three reviewers, one of whom, Volker Dötsch (Reviewer #1), is a member of our Board of Reviewing Editors.

The following individual involved in review of your submission has agreed to reveal their identity: Michael Summers (Reviewer #2).

The reviewers have discussed the reviews with one another and the Reviewing Editor has drafted this decision to help you prepare a revised submission.

Summary:

This is a very nice report of the first direct crystallographic visualization of the HIV-1 Gag CA-SP1 junction. This confirms the long-standing but unproven prediction that the region is a helix, and reveals details of its packing in the immature lattice. The crystallization apparently required heroic efforts. The study is nicely complementary to the Schur et al. cryo-EM study of the immature HIV-1 CA lattice, as the fits shown in Figure 1—figure supplement 3 look quite reasonable. The functional studies using in vitro reconstitution are quite thorough.

Essential revisions:

The following concerns should be addressed in a revised manuscript:

1) The crystals of CTD-SP1 were made up of flat hexagonal lattice (subsection “HIV-1 CA CTD-SP1 recapitulates the immature Gag lattice”), and thus "represents a flattened version of the immature HIV-1 Gag lattice". Considering this, it may not be appropriate to say CTD-SP1 hexamer and dimer give "excellent" fits to the cyroEM map of the immature HIV-1 Gag lattice (aforementioned subsection), which is a spherical shell. These Gag lattices are both made up by hexameric and dimeric interfaces, so which interface is different between the curved and flat Gag lattice? Or both of them are slightly different?

2) Related to this, do the authors have any idea of what determines the curvature of Gag lattice? Does the N-terminal domain of CA contribute to this? However, as the authors noticed, "The CTD-SP1 fragment of HIV-1 Gag contains the minimal information required to assemble the immature lattice, whereas the NTD is dispensable" (subsection “HIV-1 CA CTD-SP1 recapitulates the immature Gag lattice”). Does that mean there is certain artifact in the reported structure due to crystallization?

3) The helical structure of single SP1 was previously determined in 30% trifluoroethanol (TFE) using solution NMR (Morellet et al., Protein Sci,14:375-86, 2005), in which the helix is further extended toward the C-terminal of SP1. Do the authors know any data that support that these residues in the disordered region of the authors' crystal structure do not contribute to the assembly of immature virion? For the in vitro assembly experiment, the authors only showed the data for the residues from the half beginning of SP1 peptide (Table 2). Mutation of T371 (the first residue form the disordered region) does interfere with in vitro assembly (Table 2). Such kind of data will further validate the authors' crystal structure and their interpretations.

4) It seems pertinent to give a brief description of the helical propensity of SP1 in solution (Newman et al., Protein Sci,13:2101-7, 2004; Datta et al., J Virol, 90:1773-87, 2015) in the Introduction section, which may help to understand how the junction region is folded upon binding with RNA.

5) The file is 41 pages long, and the scientific scope seems appropriate for a full article, not a short report.

6) What was the crystallization trick that took three years to figure out?

---

## [Author Response]

*1) The crystals of CTD-SP1 were made up of flat hexagonal lattice (subsection “HIV-1 CA CTD-SP1 recapitulates the immature Gag lattice”), and thus "represents a flattened version of the immature HIV-1 Gag lattice". Considering this, it may not be appropriate to say CTD-SP1 hexamer and dimer give "excellent" fits to the cyroEM map of the immature HIV-1 Gag lattice (aforementioned subsection), which is a spherical shell. These Gag lattices are both made up by hexameric and dimeric interfaces, so which interface is different between the curved and flat Gag lattice? Or both of them are slightly different?*

2) Related to this, do the authors have any idea of what determines the curvature of Gag lattice? Does the N-terminal domain of CA contribute to this? However, as the authors noticed, "The CTD-SP1 fragment of HIV-1 Gag contains the minimal information required to assemble the immature lattice, whereas the NTD is dispensable" (subsection “HIV-1 CA CTD-SP1 recapitulates the immature Gag lattice”). Does that mean there is certain artifact in the reported structure due to crystallization?

The reviewers are correct that we should expect a difference between our structure of the flattened Gag lattice and the cryoEM structure of the curved lattice. We have revised the sentences describing the fitting and clarify that in the dimer fits, the junction helices deviate somewhat from their corresponding densities in the cryoEM map (subsection “HIV-1 CA CTD-SP1 recapitulates the immature Gag lattice”, last paragraph). We also now describe additional comparison with the PDB model by Schur et al., and our interpretation of the results (in the aforementioned subsection). In brief, we could not find a significant difference in comparing hexamers or dimers, indicating that any conformational variations that accommodate loss of curvature in our crystals are dispersed across the different subunits.

*3) The helical structure of single SP1 was previously determined in 30% trifluoroethanol (TFE) using solution NMR (Morellet et al., Protein Sci,14:375-86, 2005), in which the helix is further extended toward the C-terminal of SP1. Do the authors know any data that support that these residues in the disordered region of the authors' crystal structure do not contribute to the assembly of immature virion? For the in vitro assembly experiment, the authors only showed the data for the residues from the half beginning of SP1 peptide (Table 2). Mutation of T371 (the first residue form the disordered region) does interfere with in vitro assembly (Table 2). Such kind of data will further validate the authors' crystal structures and their interpretations.*

We have now extended our mutagenesis all the way to the end of SP1. We did find, as the reviewers anticipated, that these mutants were similar to T371A. We have added a new paragraph to the Results section to describe these mutants and our interpretation of the data, taking into account the report by Morellet et al. (subsection “The entire CA-SP1 junction is required for immature Gag assembly in vitro”, sixth paragraph).

4) It seems pertinent to give a brief description of the helical propensity of SP1 in solution (Newman et al., Protein Sci,13:2101-7, 2004; Datta et al., J Virol, 90:1773-87, 2015) in the Introduction section, which may help to understand how the junction region is folded upon binding with RNA.

We agree that these are pertinent, and have added a new paragraph to the Introduction (fourth paragraph). We also point out the significance of the helical propensity for RNA-induced folding in the Discussion (sixth paragraph).

5) The file is 41 pages long, and the scientific scope seems appropriate for a full article, not a short report.

We agree and request designation as a Research Article.

*6) What was the crystallization trick that took three years to figure out?*

There was no special trick, it simply took us 3 years to find the appropriate crystal form. We have added a few sentences to the Results section to provide more detail on the crystallization and explain why the appropriate crystals were rare (first paragraph). We also modified the relevant sentences in the Materials and methods section (subsection “X-ray crystallography”).